# Coastal Landscape Vulnerability Analysis in Eastern China—Based on Land-Use Change in Jiangsu Province

**DOI:** 10.3390/ijerph17051702

**Published:** 2020-03-05

**Authors:** Yangfan Zhou, Lijie Pu, Ming Zhu

**Affiliations:** 1School of Geography and Ocean Science, Nanjing University, Nanjing 210023, China; dg1827051@smail.nju.edu.cn (Y.Z.); zhumingnju@126.com (M.Z.); 2Key Laboratory of Coastal Zone Exploitation and Protection, Ministry of Natural Resources, Nanjing 210023, China

**Keywords:** human activities, land-use change, landscape vulnerability assessment, rapid urbanization areas

## Abstract

The unreasonable land use in rapid urbanization areas induced by large-scale urban construction activities have caused massive ecological issues. In this study, landscape vulnerability index (*LVI*) and landscape human interference index (*LHAI*) were originally addressed and calculated using multi-temporal land-use data from 2000 to 2015. Then, the spatial-temporal relationship assessment model of landscape fragility caused by human activities were constructed for each county of Jiangsu Province, China, so as to analyze the spatial distribution of landscape vulnerability and determine the impacts of artificial disturbance on landscape vulnerability. The results showed: (1) The number of counties with middle and high landscape vulnerability increased from 20 in 2000 to 27 in 2015 with a peak value (33) in 2010. (2) Counties with high-intensity human activities showed an upward trend. (3) Land use generally has a significant and diverse impact on landscape vulnerability. At the county level, the *LVI* was positively correlated with the *LHAI* before 2010 and was followed by a negative correlation of them. As concluded from this study, a total of four sub-regions (continuous benefit zones, variable benefit zones, continuous harmful zones, and variable harmful zones) have been identified for sustainable landscape management in the future. (4) The *LVI* suggests that the landscape vulnerability in Jiangsu did not continue to deteriorate in the study period. Further, accelerated land exploitation has produced a positive impact on regional economic development and ecological protection. This study provided an effective method set for analyzing the environmental impacts caused by human activities and promoting future ecosystem management in coastal areas.

## 1. Introduction

As an important part of the terrestrial ecosystem, land provides fundamental ecological services for people and animals’ survival. Because of large-scale urban construction activities in rapid urbanization areas, the contradiction between rapid economic growth and limited land resources has become increasingly serious. Landscape pattern changes caused by land use in rapid urbanization areas are a huge ecological risk, which will inevitably affect regional ecological processes, and may lead to biodiversity loss, by the manner of enhancing urban heat island effects, environmental pollution, and reducing natural landscape areas. These adverse environmental and ecological impacts of landscape changes are generally severe and even contribute to a range of public-health issues [1,2]. A recent study [3] showed that the continued expansion of a built-up area can raise temperatures and increase heat-related health risks. To coordinate land use with regional ecological security and human health, many countries have introduced the instrument of perimeters to prevent urban development. Specifically, Switzerland issued the Sectorial Plan of Cropland Protection (SPCP) and defined approximately 4400 km^2^ of fertile agricultural land, known as ‘prime cropland’, as an area to be protected permanently against urban development; the Danish Planning Act defined rural zones as priority areas for agricultural activities and established that municipal plans must identify priority areas for agricultural land use; and the Austrian Spatial Development Concept set general goals to limit urban sprawl and conserve agricultural land [4]. China has also explored the demarcation of an ‘urban development boundary’, ‘farmland protection red line’, and ‘ecological protection red line’ to alleviate greater land competition affected by rapid economic development [5].

Because of land reclamation, expansion of construction land, and other urbanization activities, ecological security has been threatened by changes in the structure and function of terrestrial ecosystems. Analyzing and predicting land-use changes at different temporal and spatial scales act as powerful tools to assess regional ecology [6]. The landscape structure is the external manifestation of different land-use patterns in the spatial combination. Different landscape indices with highly concentrated landscape pattern information can quantitatively reflect the structural composition, spatial allocation, dynamic changes, and ecological impacts of land-use landscapes.

As a significant indicator of ecological fragility, the landscape vulnerability index can be used to quantitatively analyze the sensitivity of landscape patterns to external disturbances and changes in landscape systems resulting from the lack of adaptability [7]. Ecological vulnerability assessment has become a research focus for promoting ecological civilization. However, most previous studies focused on the unstable areas that are greatly affected by natural factors, such as the Tibetan Plateau [8], the transition zone between desert and oasis [9], the ecotone of farming and husbandry [10], coastal areas [11], and wetland areas [12]. Few people assessed rapidly urbanizing areas, which are mainly affected by human activities. In addition, from the perspective of the classification of natural areas, related research always ignored administrative boundaries and used a grid scale to reveal ecological vulnerability [13]. Considering that the different changes in land use in the process of urbanization always depend on the development planning of different regions, it is necessary to study the temporal and spatial changes of ecological vulnerability at the county scale.

Jiangsu Province is one of the representative regions of rapid urbanization in China and has witnessed the contradiction between rapid economic development and limited land resources. Imbalances in the economic level and the diversity of natural resources of the counties have led to different development plans. For example, cities with higher economic levels can easily attract investment from domestic and foreign companies, thereby expanding construction land. Because of lower rents and inexpensive labor, the peripheries of metropolitan counties with lower economic levels have suffered large losses of agricultural land caused by the relocation of commercial and residential activity areas [14]. In addition, people in coastal counties have begun to promote the construction and utilization of coastal wetland areas. With economic development and urban construction, human factors have become the main driving force for the changes in and competition between land-use types [15]. More seriously, these may lead to the vulnerability of regional terrestrial ecosystems. Land use and ecological effects have obvious cumulative and regional characteristics [16].

The research objectives of this study were to (1) analyze the degree of landscape vulnerability change caused by human activities in 78 counties of Jiangsu Province from 2000 to 2015, (2) better understand the spatial evolution trajectory of the landscape vulnerability change rate in the province, and (3) determine the impacts of artificial disturbance on landscape vulnerability. To achieve the above purpose, in this paper, the following were done: (1) The spatiotemporal changes in landscape vulnerability and human activity intensity were analyzed at county scale in Jiangsu Province by combining the landscape vulnerability index (*LVI*) with the landscape human interference index (*LHAI*). (2) The change rate of *LVI* was calculated, and the center-of-gravity model was used to directly compare the degree of landscape vulnerability change in each county. The policies of counties with higher levels of vulnerability reduction were analyzed, and it was found that more attention should be paid to counties with higher levels of irrational land use. (3) The relationship assessment model was used to measure the relationship between human disturbance and landscape vulnerability. This also laid the foundation for establishing sustainable strategies to reduce land vulnerability in different counties.

## 2. Materials and Methods

### 2.1. Study Area

Jiangsu Province, belonging to Yangtze River Delta urban agglomeration of eastern China, is located within 30°45’–35°20’N and 116°18’–121°57’E, with an area of 10.72×10^4^ km^2^, including 78 county-level administrative districts (Figure 1). Due to its favorable geographical conditions and plenty of economy, Jiangsu plays an essential role in modernization and urbanization and becomes an important component of the development strategy of the ‘Yangtze River Economic Zone’.

Nevertheless, since China’s Reform and Opening up in 1978, Jiangsu has been experiencing unprecedented rates of urbanization and industrialization. Frequent changes in land use are the result of the blind expansion of urban construction land and have also led to serious ecological problems. By the end of 2015, the land development intensity of Jiangsu Province reached 21.1%, and that of Suzhou-Wuxi-Changzhou urban agglomeration was about 30%. The province’s per capita arable land was only 0.057 ha, less than 60% of the national average. The ecological space such as forests and wetlands reduced by nearly 18×10^4^ ha, resulting in the endangerment of many animals and plants, and the degradation of the ecosystem intensified [17]. Because of the unreasonable land-use structure, regional ecological problems have become increasingly prominent, regional environmental capacity has become tighter, and resource carrying capacity has continued to be weaken.

Considering the special economic contribution and acute land-use contradiction, it is of great significance to analyze the spatiotemporal characteristics of landscape vulnerability caused by land-use change in Jiangsu Province. This will help resolve land use issues, adjust the direction of land development, and achieve sustainable land-use management. In addition, as a representative province of rapid urbanization, the research results can also provide references for land-use planning, environmental management, and urbanization measures in other urbanization areas.

### 2.2. Data Source and Processing

We applied our methods using data recorded in the period between 2000 and 2015. The land-use data of Jiangsu Province was developed by the Resource and Environmental Science Data Center (RESDC) of the Chinese Academy of Sciences (http://www.resdc.cn/). It was based on the Landsat TM remote sensing data of the entire Chinese country with a spatial resolution of 30m and transformed the grid data into vector data for subsequent research. According to the land classification system of RESDC [8] and the actual situation of the study area, the land-use categories were divided into 6 types, including cultivated land (paddy fields, dry land), forest land (with forest land, shrub forest, sparse forest land, other forest land), grassland (high-coverage grassland, medium-coverage grassland, low-coverage grassland), waterbody (river canals, lakes, reservoir pits, coastal wetlands, beaches), construction land (urban land, rural residential areas, other construction sites), and unutilized land (Gobi desert, saline, marsh, bare land, bare rock texture). This area is best suited for this type of research because the diversity of land-use types brings significant spatial heterogeneity in the ecosystem. The Alberts projection and WGS1984 coordinate system were applied to the land-use data. All major analysis processes were performed using ArcGIS 10.5 (Esri, Redlands, CA, USA) and Origin 9.0 (OriginLab, Northampton, MA, USA) softwares.

### 2.3. Methodologies

Different land-use types interact in space to form heterogeneous land-use landscapes. The heterogeneity of landscape reflects the differences in land-use types and their spatial allocation and combination. Therefore, in regional ecological assessment, the landscape ecological vulnerability assessment based on land-use/land-cover change is a regional ecological assessment method based on the perspective of the spatial pattern [18]. As an indicator of highly concentrated landscape pattern information, the landscape index is suitable for quantitatively exploring the ecological impact of landscape pattern and land-use changes.

The study combining land-use change and landscape pattern will help to explore the relationship between land-use change and landscape ecological process and achieve the goal of sustainable land use [19]. Considering that the actual situations of land-use patterns in different regions of Jiangsu are different, respective *LVI* and *LHAI* values were calculated to evaluate the land-use change and its landscape ecological impact of 78 county-level areas (districts, counties, and county-level cities)(During the study period, the boundary of counties of Suzhou city was adjusted: In 2001, Wu county was abolished and divided into Wuzhong district and Xiangcheng district. In 2012, Wujiang city was abolished and changed to Wujiang district. In order to ensure the consistency of the research units, the calculation was carried out in accordance with the regional scope before the adjustment, that is, the administrative units at the county level included in Suzhou city were the Suzhou downtown, Wu county, Wujiang city, Changshu city, Zhangjiagang city, Kunshan city and Taicang city.).

#### 2.3.1. Landscape Vulnerability Index (LVI)

*LVI* represents the fragility of the internal structure of different landscapes and reflects the adaptability of landscape systems to external disturbances. Generally, the landscape sensitivity index (*LSI*) and landscape adaptation index (*LAI*) are selected to construct the *LVI* [20]. The fragility of regional landscape increases with the increase of *LVI* value.
(1)LVI=LSI×(1−LAI)

(1) Depending on the changing intensity of external influence factors and landscapes, *LSI* is composed of landscape vitiation index (*V_i_*) and landscape type interference index (*U_i_*). The formula is:(2)$$LSI=\sum_{i=1}^{n}V_{i}\times U_{i}$$
where *n* is the total number of landscape *i*. *V_i_* reflects the loss of different landscape types after the disturbance. Suggested by the study of Li et al. [21] and referencing the actual study area, there are six types of land uses in the study area, of which unutilized land is the most vulnerable, followed by water, cultivated land, grassland and forest, and construction land is the most stable land. Therefore, unutilized land is represented as level 6, and other land types decrease in order. The level values are normalized to 0~1, and the final corresponding *V_i_* values are 0.2857, 0.2381, 0.1905, 0.1429, 0.0952, and 0.0476, respectively [22]. *U_i_* refers to the landscape changes from a single, continuous, and homogeneous whole to a complex, broken, and heterogeneous mosaic under the influence of natural and human factors. It consists of landscape type fragmentation index (*F_i_*), separation index (*S_i_*), and dominance index (*D_i_*). The formulas are given as:(3)$$U_{i}=aF_{i}+bS_{i}+cD_{i}$$
(4)$$F_{i}=\frac{N_{i}}{A_{i}}$$
(5)$$S_{i}=\frac{\sqrt{DI_{i}}}{2P_{i}}$$
(6)$$D_{i}=dL_{i}+eP_{i}$$
where *DI_i_* = *N_i_/A*, *P_i_* = *A_i_/A*, *L_i_* = *N_i_/N*, *L_i_* is the relative density of the landscape, *P_i_* is the relative coverage of the landscape. Furthermore, d and e represent the weights of *L_i_* and *P_i_*, respectively, with d = 0.4, e = 0.6. *DI_i_* is the landscape distance index, *N_i_* is the patch of the *i*th landscape, *N* is the total number of landscape patches, *A_i_* is the area of the *i*th landscape, and *A* is the total area. Further, a, b, and c represent the weights of *F_i_*, *S_i_*, and *D_i_* respectively, and a + b + c = 1. According to the previous study [23], a = 0.5, b = 0.3, c = 0.2 (a = 0.2, b = 0.3, c = 0.5 for unutilized land [22]).

(2) *LAI* refers to the ability of the landscape to adapt and self-recover under external disturbances, which consists of plaque rich density index (*PRD*) [24], Shannon diversity index (*SHDI*) [25], and Shannon uniformity index (*SHEI*) [25]. Combined with the detailed introduction of the three indices in the references, the specific calculation process can be completed in ArcGIS10.5.
(7)LAI=PRD×SHDI×SHEI

#### 2.3.2. Landscape Human Interference Index (LHAI)

The landscape changes in the study area are affected by natural and human factors. The former plays an important role on the long-term scale, while human factors in the short term [26]. From 2000 to 2015, human activities were the dominant factor in the landscape evolution of Jiangsu. Natural features were inversely proportional to the artificial features of the landscape. *LHAI* can effectively reflect the intensity of regional landscapes affected by human activities [27] and quantitatively describe changes in land use. The index can be calculated by:(8)$$LHAI=\sum_{i=1}^{N}A_{i}*R_{i}/A$$
where *R_i_* represents the factor of impact on resources and environment of the *i*th landscape type, *A_i_* is the area of the *i*th landscape, and *A* is the total area. The specific value of different landscapes in Jiangsu was determined by the relevant studies (Table 1) [22,28].

#### 2.3.3. Spatial Gravity Center Model

By introducing the concept of the regional gravity center, the following spatial gravity center model of *LVI* change rate was established. The spatial gravity center model has a unique advantage in exploring the spatial-temporal changes of factors. It can intuitively and accurately reveal the distribution and evolution characteristics of these factors in two-dimensional space.

The landscape vulnerability in each county has changed over time. The positive change rate in some counties indicates that the landscape vulnerability is continuously deepening, while the negative rate indicates that the vulnerability has been effectively alleviated. The trajectory of the gravity center of the county with a positive change rate can reflect which areas are more ecologically friendly to land use. The trajectory of the center of gravity of the areas with a negative change rate can reflect the need for land-use reform in these areas.

If the study area consists of *n* units and (*x_i_*, y*_i_*) is the geometric coordinate of *i*th unit (*i* = 1, 2, 3, …, n), the factor’s gravity center coordinate ($\bar{x}$, $\bar{y}$) can be represented by:(9)$$\bar{x}=\frac{\sum_{i=1}^{n}x_{i}\times m_{i}}{\sum_{i=1}^{n}m_{i}}$$
(10)$$\bar{y}=\frac{\sum_{i=1}^{n}y_{i}\times m_{i}}{\sum_{i=1}^{n}m_{i}}$$
where *m_i_* refers to the attribute value of the *i*th unit.

Based on the analysis in Section 2.3.1, we calculated the gravity center coordinates of the county with a positive *LVI* change rate and the county with a negative *LVI* change rate to analyze the change trajectories of these gravity centers.

The moving direction of the gravity center can be calculated by:(11)$$\alpha =\left[\frac{k\times \pi }{2}+arctan\left(\frac{{\bar{y}}_{t_{2}}-{\bar{y}}_{t_{1}}}{{\bar{x}}_{t_{2}}-{\bar{x}}_{t_{1}}}\right)\right]\times \frac{180{}^{\circ}}{\pi }$$
where  α  represents the deviation angle of the gravity center during the study period, (${\bar{x}}_{t_{1}}$*,*$\;{\bar{y}}_{t_{1}}$) and (${\bar{x}}_{t_{2}}$*,*$\;{\bar{y}}_{t_{2}}$) refer, respectively, to the gravity center coordinate at the beginning and end of the study, *t*_1_ and *t*_2_ represents, respectively, the beginning and end of study period, *k* refers to the adjustment coefficient to make sure that  α∈(−180°,180°)  and *k* = 0, 1 and 2, π refers to the Ludolph’s coefficient. This paper defines the anti-clockwise direction as the positive direction, and the east is 0°.

The moving distance of the gravity center can be calculated by:(12)$$L=\sqrt{{\left({\bar{x}}_{t_{2}}-{\bar{x}}_{t_{1}}\right)}^{2}+{\left({\bar{y}}_{t_{2}}-{\bar{y}}_{t_{1}}\right)}^{2}}$$
where *L* refers to the moving distance of the gravity center during the study period, (${\bar{x}}_{t_{1}}$,$\;{\bar{y}}_{t_{1}}$) and (${\bar{x}}_{t_{2}}$,$\;{\bar{y}}_{t_{2}}$) refer, respectively, to the gravity center coordinate at the beginning and end of the study, *t*_1_ and *t*_2_ represents, respectively, the beginning and end of the study period.

#### 2.3.4. Relationship Analysis

To evaluate the impact of *LHAI* on *LVI*, a relationship assessment model was established to calculate the relationship between *LVI* and *LHAI*. Relationship analysis is a qualitative analysis method used to study the influence of one changing factor on the other or a set of key indicators. It aims to explain the changes in key indicators that are caused by changes in a factor to investigate the influence relationship between them [29]. The formula is given as:(13)$$\beta =\frac{\left[\left(LVI_{t}-LVI_{t_{1}}\right)/LVI_{t_{1}}\right]}{\left[\left(LHAI_{t}-LHAI_{t_{1}}\right)/LHAI_{t_{1}}\right]}$$
where  β represents the value of relation, $LVI_{t_{1}}$ and $LVI_{t}$ refer to, respectively, *LVI* at the start and end of the study period, and $LHAI_{t_{1}}$ and $LHAI_{t}$ refer to, respectively, *LHAI* at the start and end of study period. If β < 0, there is an inverse change in the relationship between *LVI* and *LHAI*. If β > 0, the two indexes are positively correlated [30].

## 3. Results and Discussion

### 3.1. Temporal and Spatial Changes in Landscape Vulnerability

#### 3.1.1. Geospatial Features

Observing and analyzing natural complexes from a regional perspective is the conclusion of a comprehensive study of processes and types in geography. Discussing the spatial changes of landscape fragility in different counties is helpful to assess the impact of land-use change on ecological vulnerability.

(1) Spatial distribution

By using natural break point and a relative index method (at an interval of 0.02) [8,31], the study area can be classified into low- (*LVI* < 0.06), middle- (0.06 ≤ *LVI* ≤ 0.08), and high-vulnerability zones (*LVI* > 0.08) (Figure 2).

Because of the rapid economic development and frequent landscape reconstruction, Jiangsu experienced the expansion of the middle- and high-vulnerability areas from 2000 to 2010, especially in Binhai and Sheyang, where coastal wetland exploitation increased significantly. The biggest threat to coastal wetlands is the coastal reclamation projects, leading to large-scale losses of ecological function.

In the world, especially in developing regions, human activities are the main driving force in the evolution of coastal wetlands, and natural wetlands are gradually being transformed into artificial areas [32]. Similarly, this change has occurred in other parts of China, and Gu et al. [33] analyzed the land transformation in Jiaozhou Bay between 1952 and 2002 and found that approximately 33.7% of the wetlands were converted to artificial wetlands. In the southern counties, Nanjing City, as the capital of Jiangsu, suffered enhanced landscape reconstruction in the process of rapid urbanization [34].

Later, because of concerns about ecological protection, the development mode of southern Jiangsu was changed to reduce the vulnerability of the landscape. Northern Jiangsu, including Suining, Pizhou, and Ganyu, relied on the advantages of agricultural and sideline products and forest resources to develop low-level industry and agriculture, and it witnessed severe soil erosion and inefficient resource use.

(2) Change in spatial gravity center

Gravity center analysis aims to explore the spatial equilibrium and mobility of landscape vulnerability in the region [32]. The *LVI* change rate of each county was chosen for the gravity center analysis to more accurately explore the spatial changes in the degree of landscape vulnerability change in the study area.

The change trajectories of gravity center of *LVI* change rates are shown in Figure 3, and the moving distances and directions of gravity centers were calculated, as shown in Table 2. 

During the study period, the gravity centers of the counties where the *LVI* change rate was positive (*LVI* was increasing) were all in northern Jiangsu. The specific location was moved from Huaian City to Donghai. Nevertheless, the gravity center of the counties where the *LVI* change rate was negative (*LVI* was decreasing) shifted from Rugao to its northwest. From an overall perspective, the gravity center of the positive *LVI* change rate moved 147.69°and 109.95 km in the northwest, while the gravity center of the negative *LVI* change rate moved 156.15°and 87.60 km in the northwest. These change trajectories illustrated that the economic and social development of eastern and southern Jiangsu was generally better than those of western and northern regions. When economic expectations are met, the ecological environment becomes the focus of urban construction. Therefore, the gravity centers of positive and negative *LVI* change rates showed a trend from the southeast coast to the northwest inland, but the exact location of the positive *LVI* change rate was located to the north of the negative *LVI* change rate.

Previously, research on gravity centers mainly focused on gross domestic product, carbon dioxide emissions, overall population, and urban population [35]. Fewer studies on landscape vulnerability have been done. The above results remind counties in northern Jiangsu to strengthen ecological protection and pursue ecological benefits, and counties in the south should maintain a green development path, minimize the negative impact of land use, and maintain a downward trend in landscape vulnerability.

#### 3.1.2. Trend of Time Variation

During 2000–2010, the number of counties with low vulnerability in Jiangsu decreased from 58 to 45. The number of middle- and high-vulnerability counties increased dramatically, with a growth rate of 65%, reflecting the deterioration of the province’s landscape vulnerability. During the following five years, the number of counties with low vulnerability increased from 45 to 51, with a growth rate of 13.3%. The number of middle- and high-vulnerability counties decreased. The province’s overall landscape vulnerability improved (Figure 4).

### 3.2. Temporal and Spatial Changes in Human Interference Intensity

#### 3.2.1. Geospatial Features

(1) Trends in spatial variation

By using natural break point and a relative index method (at an interval of 0.05) [8,31], the *LHAI* in Jiangsu was divided into three levels, namely, the low- (*LHAI* < 0.38), middle- (0.38 ≤ *LHAI* ≤ 0.43), and high-interference area (*LHAI* > 0.43) (Figure 5).

Overall, from 2000 to 2015, the intensity of human interference increased in Jiangsu. Two centers of high-interference areas were formed. Northern Jiangsu was concentrated in downtown Suqian, Siyang, downtown Huaian, and Lianshui, while southern Jiangsu was concentrated in downtown Suzhou, downtown Wuxi, and downtown Nanjing. The northern region comprises important agricultural production areas. As a result, because of frequent changes in cultivated land, landscape interferences in the region continuously increased. Rural construction in northern Jiangsu was characterized by a large land area, scattered layout, and inefficient use of rural residential areas. By the end of 2004, the per capita living area of rural areas in northern Jiangsu was 201.09 m^2^, far exceeding the reasonable standard of 100 m^2^ determined by the province, resulting in a high *LHAI* in rural residential areas. Relying on geographical and policy advantages, the development of export-oriented industries and intensive construction on limited land resources caused serious human intervention in the southern region. The results of the *LHAI* from 2000 to 2015 confirmed that urbanization had a serious impact on land-use change in Jiangsu. This was confirmed by many studies—for instance, a study conducted by Sun et al. [36] showed that Wuhan’s built-up areas expanded at an average rate of 4.36% during 1991–2013, resulting in a severe transformation of the living environment in urban areas.

#### 3.2.2. Situation of Time Change

As shown in Figure 6, the number of low-interference counties in the study area gradually decreased from 56 to 34 during 2000–2015. The number of middle- and high-interference counties increased gradually, and the number of high-value areas increased by 144%. Frequent land reconstruction activities led to severe artificial disturbance of the landscape.

### 3.3. Impact of Land-Use Change on Landscape Vulnerability

#### 3.3.1. County Scale

From 2000 to 2015, the number of counties with middle and high landscape vulnerability in Jiangsu increased first and then decreased, reaching the highest value in 2010 (Figure 4). During 2000–2015, the number of counties with middle and high human interference intensity increased steadily, reflecting the deepening artificial disturbance intensity of the landscape at the county scale (Figure 6).

For a comprehensive analysis, the number of counties with different β was calculated, as shown in Table 3. Between 2000 and 2010, more than 57.6% of counties had β > 0, which indicated that *LVI* was positively correlated with *LHAI* at the county scale. This can be explained by the main purpose of land-use change during this period of economic development. Because the waste of resources and environmental pollution were not considered, the original ecological pattern of the region was destroyed. After 2010, national policies, such as sustainable resource development and ecological civilization construction, were introduced. Considering the shortage of regional land resources and the limitation of environmental restoration capacity, the main purpose of land transformation was to seek a win–win model of common economic and environmental development. Therefore, the number of counties with β < 0 increased to 71.79%, which reflects that the relationship between *LVI* and *LHAI* changes inversely at the county scale.

Various assessment models have been widely used to assess the ecological impacts of land use in urbanization areas. Previous studies have analyzed the effects on ecological water quality [37], net primary productivity [38], dust emissions [39], and ecological functional vulnerability [40]. These evaluation models provide an important reference for regional sustainable land management. Compared with models that require a higher mathematical statistics basis, landscape indices are relatively simple and convenient to apply. With the calculation of *LVI* and *LHAI* in 78 counties from 2000 to 2015, it was found that the ecological environment in Jiangsu Province had not been deteriorating. Shen [41] used a land ecological security early-warning model to monitor the degree of land ecological security early warning in Jiangsu, indicating that the land ecological security index of Jiangsu Province was moving toward a better situation. This is very similar to the results of this study.

#### 3.3.2. Province Scale

The growth rate of landscape vulnerability in Jiangsu increased from 6.28% to 24.88%. Meanwhile, the province scale was consistent with the trend of the county scale from 2000 to 2010 but was opposite during 2010–2015 (Table 4). Because the landscape index was sensitive to the research scale [13], although the landscape vulnerability at the county scale decreased from 2010 to 2015, the extent and intensity of weakened areas were still smaller than the enhanced area at the province scale. Although the landscape fragility increased, the growth rate showed a stable and declining trend. This indicated that the overall trend of land use in the province tended to be eco-friendly. To promote the sustainable use of land resources comprehensively, it is necessary to deepen the understanding of the relationship between ecological environment protection and land development and to improve the relationship between market driving and government guidance.

During the study period, the *LHAI* at the province scale increased, which was consistent with the trend of landscape vulnerability. It showed that the intensity of land use in Jiangsu has been continuously increasing in the past 15 years. Although this promoted economic development and enriched social infrastructure construction, there was a negative impact on the regional ecological environment. In the future, the adjustment of land-use structure should promote the comprehensive development of ecology and economy.

### 3.4. Spatial Planning and Causes of the Relationship between LVI and LHAI

As described in Section 2.3.4, the symbol β can be used to describe the relationship between *LVI* and *LHAI*. If β > 0, *LVI* is positively correlated to *LHAI*, indicating that human activities will increase landscape vulnerability; that is, human activities are harmful to the environment. If β < 0, *LVI* is inversely proportional to *LHAI*, indicating that human activities will reduce landscape vulnerability; that is, human activities are beneficial to the environment. Considering β > 0 or β < 0 during 2000–2005, 2005–2010, and 2010–2015, a classification scheme was established, as shown in Table 5. The province can be divided into four areas based on different characteristics (Figure 7).

The spatial distribution results showed that 71.8% of the counties in Jiangsu were located in continuous benefit zones and variable benefit zones caused by human activities. It is obvious that human factors helped to alleviate the province’s landscape vulnerability. With the support of local economic and environmentally friendly policies, represented by Nanjing City, Zhenjiang City, Changzhou City, Wuxi City, and Taizhou City, the southern and central part of Jiangsu adhered to ecological priority, actively developed urban artificial green spaces and green protective belts, and built urban ecological spaces, promoting the integration of ecological, agricultural, and urban spaces [17]. The overall vulnerability of the landscape weakened, which fully proved that human intervention was not an absolute negative factor in protecting landscape ecology [42]. 

However, with the rapid development of the economy, the areas represented by Wujiang, Wuxian, and Zhangjiagang also became more vulnerable, owing to the frequent conversion of agricultural land to construction land according to the expansion of downtown Suzhou. In the future, they should change the mode of agricultural development and stop the conversion of farmland to construction land caused by urban expansion. Combining regional advantages, they are better at developing suburban-sightseeing agriculture to increase artificial landscape comfort and enhance ecological value, so that they can carry out land-scale operations to promote agricultural transformation and upgrading.

Represented by most counties in Xuzhou City, Liangyungang City, and Yancheng City, northern Jiangsu experienced ecological problems affected by human activities. Nonetheless, the causes were not similar in these areas.

Xuzhou City was an important agricultural area with superior soil quality, but agricultural land plots were fragmented, the pace of eco-agricultural transformation was slow, and the structure of agricultural land was inefficient. There was thus an urgent need for transformation in agricultural production. Previous studies demonstrated that the fragmentation of agricultural land may lead to a decline in agricultural product quality [43] and land degradation [44], and it may also endanger food security [45]. Accordingly, for counties with agriculture as the leading industry, large-scale farmland management is an effective measure to ensure ecological security. In addition, new agricultural development models, such as the construction of agricultural ecological parks and plant factories, are also worth promoting.

Based on the beautiful natural landscape, Lianyungang City improved the economic and social life infrastructure with the help of tourism. However, environmental problems have also been caused by tourists’ interference. As confirmed by Chen and Bian [46], the irrational use of resources and increased land demand for recreational facilities and hotels has led to deterioration of the surrounding environment, such as increased accumulation of solid waste and levels of air and water pollution. Armono et al. [47] also proved that an increase in the number of tourists leads to the degradation of coastal resources and the reduction of ecosystem services in a case study of the ecotourism capacity of Barulan National Park, Indonesia. Therefore, tourist cities in the province should pay attention to protecting the original wetlands and lakes and try to avoid large-scale introduction of commercial, road, and accommodation construction land to reduce the pressure on the environment of the scenic spot.

Furthermore, Yancheng City’s active development of industry and agriculture in coastal wetland development zones is responsible for the increased vulnerability of the landscape. Wang et al. [48] analyzed the development and management of land reclamation in China and found that land reclamation had also become an important way for China to meet growing residential space and development needs, such as in Jiangsu, Shanghai, Zhejiang, and Fujian Provinces. The results of this study also suggest that rapid urbanization has simultaneously induced many adverse impacts on the coastal wetland environment [49], not just in urban areas [50]. To solve the fundamental contradiction between economic development and ecosystem protection, areas similar to Yancheng City should follow the concept of sustainable development, strengthen laws and regulations, improve land space planning, fully evaluate the negative impact of coastal wetland development, and enhance public involvement in reclamation management.

Only by combining the location characteristics and resource advantages of each county and rationally formulating the development methods of land resources can the speed of economic development be accelerated and regional ecological benefits be improved without causing unnecessary damage to the ecological environment.

## 4. Conclusions

This study used the county scale as the unit of analysis. Based on the land-use data of Jiangsu Province from 2000 to 2015, *LVI* and *LHAI* were constructed to analyze quantitatively the impact of regional land-use changes on the ecological environment.

In summary, the *LVI* of Jiangsu Province first increased and then decreased, reaching its maximum value in 2010, when the growth rate of middle- and high-vulnerability counties was 65%. During the study period, the *LHAI* increased steadily. At the county scale, land-use change in Jiangsu had a significant impact on the ecological environment. From 2000 to 2010, more than 57.6% of counties suffered from landscape vulnerability, and the intensity of human disturbance was positively correlated. During 2010–2015, the two indices were negatively correlated in more than 71.79% of counties, which reflects that, under the strategic background of national ecological security in recent years, human activities affecting the landscape had changed from destruction to maintenance. Finally, according to the relationship between *LVI* and *LHAI* of each county, Jiangsu Province could be distributed into continuous benefit zones, variable benefit zones, continuous harmful zones, and variable harmful zones caused by human activities. Despite the results achieved, more analysis needs to be conducted. Because there are no more land-use data, it was not possible to use multiple periods of land-use data to evaluate quantitatively the relationship between *LVI* and *LHAI*. In addition, if the length of the article is not a limitation, more spatial analysis methods (such as the Moran index spatial analysis) should be added to analyze the landscape vulnerability caused by human activities.

Under the United Nations 2030 sustainable development framework, sustainable land use has become a focus of global attention. Jiangsu Province also faces a trade-off between land development and ecological protection. The results of this study showed that, from the perspective of landscape ecology, the method of calculating *LVI* and *LHAI* to study the impact of land-use change on landscape vulnerability was simple and effective. The results clarify the relationship between landscape pattern changes and ecological processes and enrich the body of research on land-use change. The study also proves that it is effective to apply the landscape index method to analyze the ecological impact caused by land-use change. According to the assessment results, each region can accurately assess the landscape ecological impacts of local land-use changes and even simulate and determine the ecological impacts of planning schemes at an early stage to adjust in time and avoid increasing the vulnerability of landscapes. In addition, for land planners, these methods can be crucial for identifying the impacts of their decisions on landscape transformation. This can help define land priorities and design new policies to avoid undesirable ecological impacts in the future.

## Figures and Tables

**Figure 1 ijerph-17-01702-f001:**
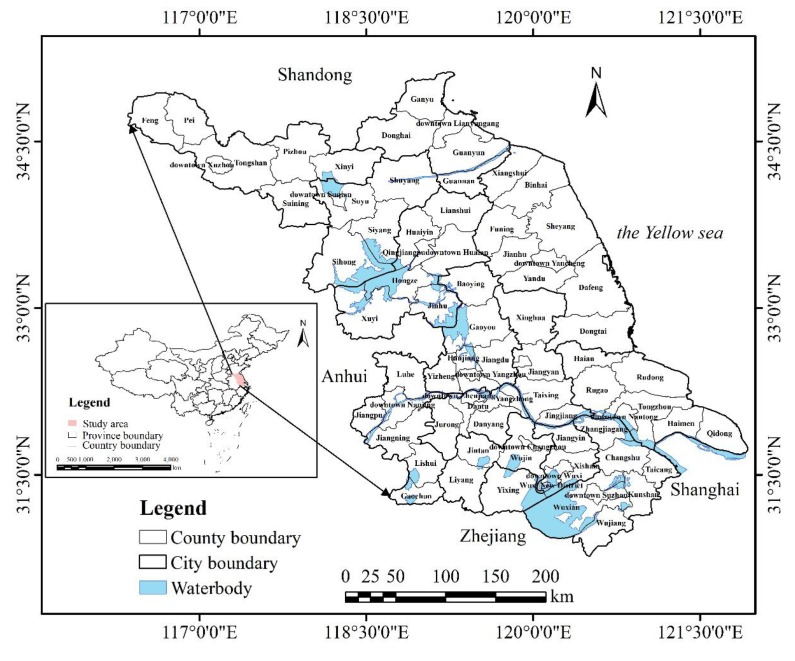
Location of study area in eastern China and 78 counties in the region.

**Figure 2 ijerph-17-01702-f002:**
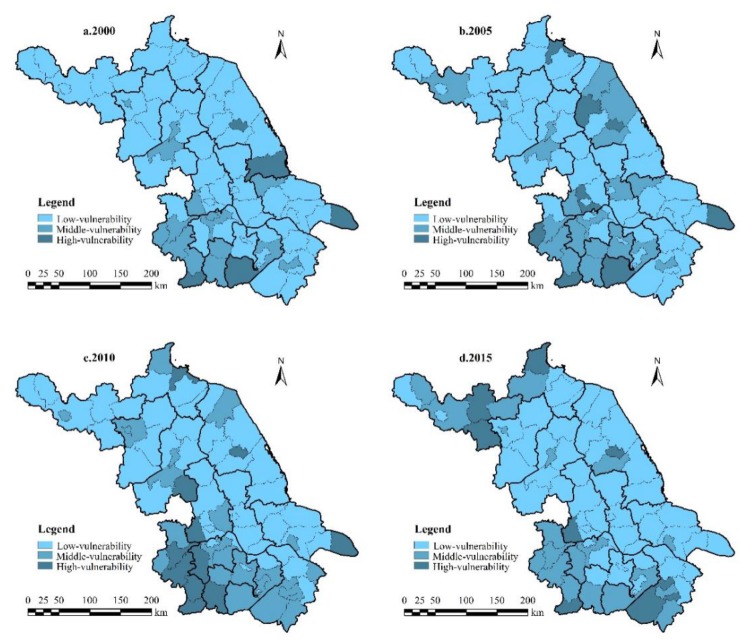
Spatial distribution of *LVI* in 2000, 2005, 2010, and 2015.

**Figure 3 ijerph-17-01702-f003:**
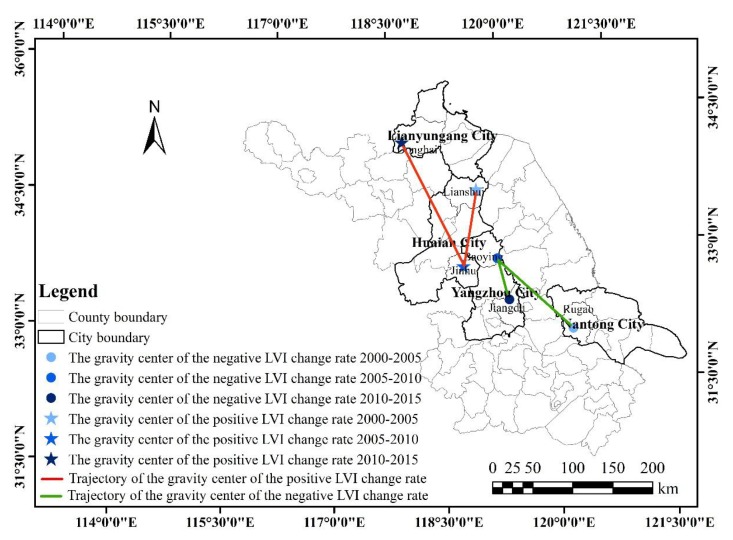
The trajectories of gravity centers of positive and negative landscape vulnerability index *(LVI)* change rates.

**Figure 4 ijerph-17-01702-f004:**
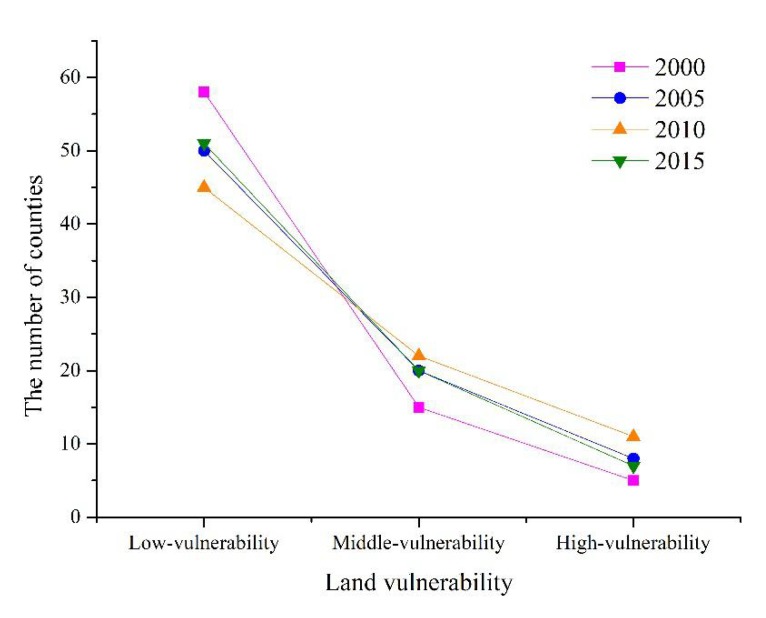
The number of counties with different *LVI.*

**Figure 5 ijerph-17-01702-f005:**
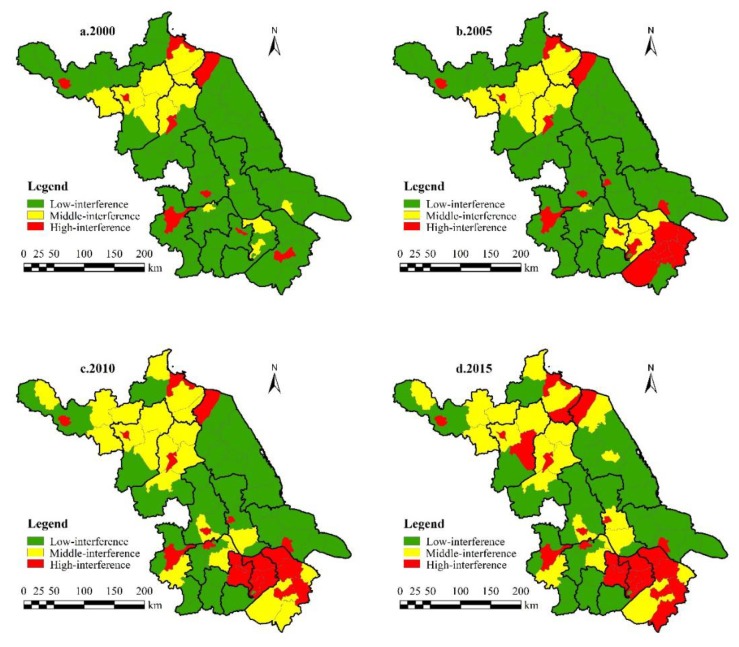
Spatial distribution of land human interference index (*LHAI)* in 2000, 2005, 2010, and 2015.

**Figure 6 ijerph-17-01702-f006:**
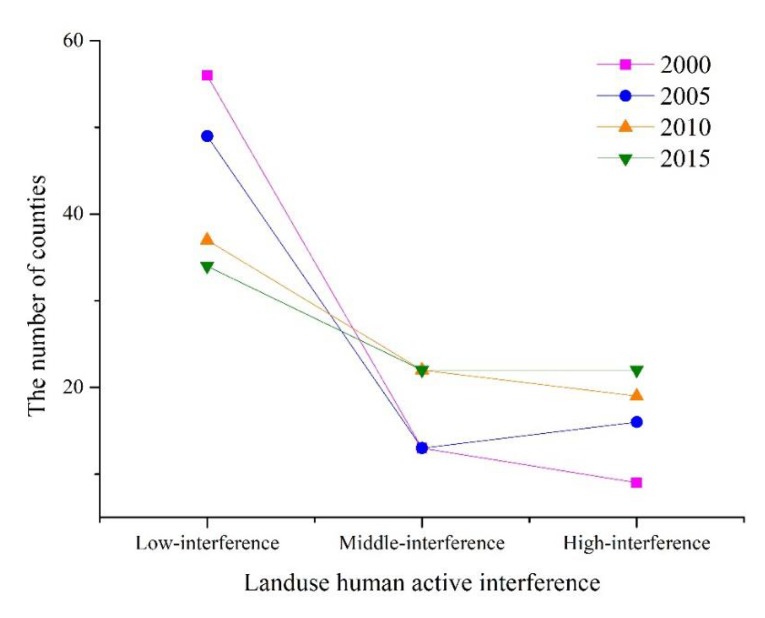
The number of counties with different *LHAI.*

**Figure 7 ijerph-17-01702-f007:**
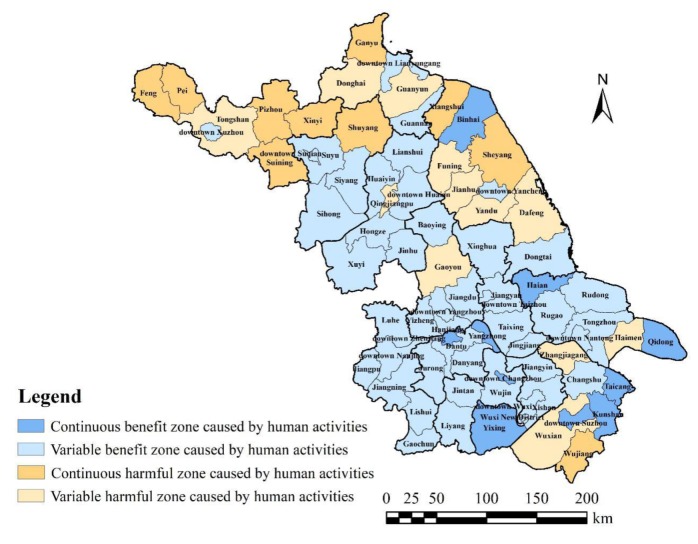
The result of spatial distribution according to the relationship between *LVI* and *LHAI.*

**Table 1 ijerph-17-01702-t001:** Impact factors on landscape resources and environment.

Landscape Type	Status	Impact Factor
Cultivated land	Its impact on resources and the environment is small, and some of them are reversible	0.25
Forest land	It has the function of ecological maintenance and has little impact on resources and environment	0.1
Grassland	It has the function of ecological maintenance and has little impact on resources and environment	0.1
Waterbody	Most of the waterbodies belonging to the study area are used for aquaculture and traffic transportation, which are greatly affected by humans and have irreversible impacts on resources and the environment	0.37
Construction land	It is greatly affected by human activities, most of which are irreversible and have a significant impact on resources and the environment	0.85
Unutilized land	It has a slight impact on resources and the environment, most of which are irreversible	0.48

**Table 2 ijerph-17-01702-t002:** Moving distances and directions of gravity centers.

Gravity Center	Moving Direction (°)	Moving Distance (km)
Positive *LVI* change rate during 2000–2005		
Positive *LVI* change rate during 2005–2010	−99.21933028	97.06707694
Positive *LVI* change rate during 2010–2015	116.5900208	172.8695394
Negative *LVI* change rate during 2000–2005		
Negative *LVI* change rate during 2005–2010	137.606495	128.9835979
Negative *LVI* change rate during 2010–2015	−73.63187194	53.72771954

**Table 3 ijerph-17-01702-t003:** The number of counties with different β.

Year	The Number of Counties with β < 0	The Number of Counties with β > 0
2000–2005	33	45
2005–2010	33	45
2010–2015	56	22

**Table 4 ijerph-17-01702-t004:** *LVI/LHAI* and their change rates at province scale.

Year	*LVI*	The Change Rate of *LVI*	*LHAI*	The Change Rate of *LHAI*
2000	0.054399		0.344772	
2005	0.057817	6.282839%	0.363015	5.291249%
2010	0.072392	25.210100%	0.372573	2.632977%
2015	0.090406	24.883740%	0.389341	4.500529%

**Table 5 ijerph-17-01702-t005:** Classification method according to the relationship between *LVI* and *LHAI.*

β During 2000–2005	β During 2005–2010	β During 2010–2015	Significance	Name	The Number of Counties
β < 0	β < 0	β < 0	Human activities are always beneficial to the environment	Continuous benefit zone caused by human activities	10
β > 0	β > 0	β < 0	Human activities have not always been beneficial to the environment, but recently they are good for the environment	Variable benefit zone caused by human activities	46
β > 0	β < 0	β < 0			
β < 0	β > 0	β < 0			
β > 0	β > 0	β > 0	Human activities are always harmful to the environment	Continuous harmful zone caused by human activities	10
β < 0	β > 0	β > 0	Human activities have not always been harmful to the environment, but recently they are bad for the environment	Variable harmful zone caused by human activities	12
β > 0	β < 0	β > 0			
β < 0	β < 0	β > 0			

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
