# Peer review of "Coastal Landscape Vulnerability Analysis in Eastern China—Based on Land-Use Change in Jiangsu Province"

_ijerph, 2020, doi:10.3390/ijerph17051702_

Round 1
Reviewer 1 Report
While the presented methodology is not new (for example, the chosen metrics are pre-existing), it is a nice application that tracks data over the years and shows the impact of proactive government policies in moving a region towards sustainability.
It would have been nice to see a model in this paper, but the authors have shifted that to future research.
The colored dots in Figure 3 are difficult to see on the map.
The summary and analysis in section 3.4 are nicely done.
The language needs editing in general.
Author Response
Response to Reviewer 1 Comments
Dear Editors and Reviewers:
Thanks for your letter concerning our manuscript entitled “Coastal Landscape Vulnerability Analysis in Eastern China — Based on Land-Use Change in Jiangsu Province” (ID: ijerph-704770). Those comments are helpful for revising and improving our paper, as well as the important guiding significance to our researches. We have studied comments carefully and made correction which we hope meet with approval. Revised portion are marked in red in the paper. The main corrections in the paper and the responds to the reviewers’ comments are listed below point by point.
Authors’ Responses to Reviewer’s Comments (Reviewer 1)
Author’s Notes
Reviewer 1:
While the presented methodology is not new (for example, the chosen metrics are pre-existing), it is a nice application that tracks data over the years and shows the impact of proactive government policies in moving a region towards sustainability.
Point 1: It would have been nice to see a model in this paper, but the authors have shifted that to future research.
Response 1: Affected by the amount of the land-use data and limited by the length of the article, we hope to establish more quantitative models in the future. We have explained these in results and the discussion (line 457-461).
Point 2: The colored dots in Figure 3 are difficult to see on the map.
Response 2: We have revised the Figure 3 (line 282).
Point 3: The summary and analysis in section 3.4 are nicely done.
Response 3: Thanks for your appreciation.
Point 4: The language needs editing in general.
Response 4: We have revised the language expression by professional editing agency.
Special thanks to you for your good comments.
We appreciate for Editors/Reviewers’ warm work earnestly, and hope that the correction will meet with approval.
Once again, thank you very much for your comments and suggestions.

Reviewer 2 Report
Dear authors,
I find your study to be engaging and providing interesting methods for the analysis of landscape dynamic in the context of rapid urbanization. While the topic is not new, the novelty of your paper could come from the proposed assessment methods. However, I consider that some improvements should be done and some issues should be addressed for the paper to be worthy for publishing in IJERPH. I am looking forward to see your work published.
General comments
First of all I would kindly advice to improve the quality of the English language of the submitted paper. There are not as many writing errors as more phrasing errors. You should have a native speaker look upon your text and revise it. The current text does not provide a smooth and coherent reading experience.
Second of all, while the topic of your study slightly fits in the journal's aims and scope, I would recommend to insert in the text a better description of how the quality of ecosystems is linked with quality of life, mainly in highly urbanized regions. I suggest emphasizing how landscape dynamics could impact human health status, and why it is important to maintain a balance between natural and constructed areas. Of course, my suggestion implies the provision of a brief argumentation to increase the suitability of your paper within the journal's aims and scope.
Further I will provide specific comments on the manuscript's section and text:
Abstract: I recommend rethinking the information you wish to provide in the abstract. This section is very important as it makes a reader reading the whole paper. Thus, I suggest to provide brief descriptions of the methods used and a glimpse to the major finding or findings of your paper. Also, in the end of your abstract you should provide an argument backing the originality and applicability provided by your study.
Lines 14-15: The word "study" repeats and it alters the coherence of the sentence. I suggest: "[...] the analysis focused on Jiangsu Province in Eastern China as a case study are, [...]"
Introduction: I suggest to clearly emphasize which are the study's main objectives at the end of this section. After that you should describe the methodologies and results in relation with these objectives. Otherwise the paper lacks a clear focus.
Materials and Methods - Data source and processing:
Line 99: I suggest: "We have applied our methods using data recorded in the period between 2000 and 2015"
Line 103: Citation needed. Some readers may not know which is the land-use standard classification system.
Line 112: I suggest: "[...] using ArcGIS 10.5 and Origin 9.0. softwares"
2.3. Methodologies: I recommend that for each index to clearly specify which are the required input data. In this way the methods are replicable by other researchers in their own case studies, fact that may increase your citation numbers and the journal's visibility.
Lines 114-115: Unclear, please consider rephrasing it to provide more clarity to the sentence.
Lines 118-121: Unclear, please consider rephrasing it to provide more clarity to the sentence.
Lines 122-129: I suggest to reconsider the argumentation within this paragraph. In its current form it is unclear for the reader. Try using briefer and shorter sentences. I understand that you have calculated LVI and LHAI indexes at a county level, meaning you have 78 LVI and 78 LHAI values. Is that right?
2.3.1. Landscape Vulnerability Index
Line 142: I suggest :"land uses" to avoid repeating the word "types".
Line 158: Which study?
Line 163: How are PRD, SHDI and SHEI established? I reckon that they are calculated using spatial analysis tools and GIS processing. Right? However, a short description is required, to proper disseminate the methods to a wider range of readers.
Line 165: the word "landscape" implies the fact that it refers to land, thus it is redundant to say "land landscape"
Line 169: The same as mentioned before
Line 171: It is not clear what Aj and A stand for?
Table 1: These values were set empirical by the authors? If so, than you should provide more information on how the values were set, otherwise they lack the necessary scientific ground.
2.3.3. Spatial Gravity center model: Without trying to be rude, I can't find any relevance of this analysis within your study. I'm not saying it is wrong but "cui prodest"? I understand that you wanted to emphasize how the pressure on landscape quality moved throughout the analyzed period with the region, but this can be observed from the maps as well. Also the way the methods and results are exposed are really fuzzy. If you wish to keep this analysis in your paper, I recommend providing more arguments to highlight why it is useful and what it shows. Link it to a paper objective (which you don't have at the moment)
Results-Figure 1: I recommend changing the colors used in this maps. You used different shades of green for low vulnerability and high vulnerability which may be confusing. I recommend using different shades of the same color for your scale, with an increasing tone from low to high.
Lines 242-245: Please consider to divide the phrase into shorter and briefer sentences as in the current form it is hard to comprehend.
Figure 3: It is quite hard to read the map in its current form. It is hard to distinguish the circles from squares as they are too small and the colors hard to be identified. I recommend using larger symbols and different colors. Besides that, the map contains items not explained in the legend, such as the arrows. What does the colors represent? or the direction?
Line 291: You should cite a figure before it is exposed to the reader. In your case is the other way around. Please address this issue.
Line 295: Redundant. Please keep on "landscape" or "land use"
Figure 6 and Table 3 must be revealed to the reader before passing to other subsections.
Figure 6: The same issues emphasized for Figure 5 are occurring here. Please address them.
Line 354: The Journal's recommendation, as well as the scientific writing customs indicates that a figure should be inserted after the ending of the paragraph in which it was first cited. It is not the case for your paper. Please address this issue and follow the above mentioned recommendations.
Table 6: It is not clear, or the authors failed to provide sufficient information about what type of correlation was used for this analysis.
References: I have noticed that only 24% of your reference (24.48% more exactly) are not Chinese published papers, and from those, only one reference is from the last 5 years. While I personally acknowledge and appreciate the contribution of Chinese researchers within the international scientific community, I don't think that topics such as landscape dynamics are not tackled by others as well. The content of your references makes me wonder if your paper is not better suited for a Chinese national journal rather than an international one. Please address this issue.

Author Response
Response to Reviewer 2 Comments
Dear Editors and Reviewers:
Thanks for your letter concerning our manuscript entitled “Coastal Landscape Vulnerability Analysis in Eastern China — Based on Land-Use Change in Jiangsu Province” (ID: ijerph-704770). Those comments are helpful for revising and improving our paper, as well as the important guiding significance to our researches. We have studied comments carefully and made correction which we hope meet with approval. Revised portion are marked in red in the paper. The main corrections in the paper and the responds to the reviewers’ comments are listed below point by point.
Authors’ Responses to Reviewer’s Comments (Reviewer 2)
Author’s Notes
Reviewer 2:
Dear authors,
I find your study to be engaging and providing interesting methods for the analysis of landscape dynamic in the context of rapid urbanization. While the topic is not new, the novelty of your paper could come from the proposed assessment methods. However, I consider that some improvements should be done and some issues should be addressed for the paper to be worthy for publishing in IJERPH. I am looking forward to see your work published.
General comments
Point 1: First of all I would kindly advice to improve the quality of the English language of the submitted paper. There are not as many writing errors as more phrasing errors. You should have a native speaker look upon your text and revise it. The current text does not provide a smooth and coherent reading experience.
Response 1: We have revised the language expression by professional editing agency.
Point 2: Second of all, while the topic of your study slightly fits in the journal's aims and scope, I would recommend to insert in the text a better description of how the quality of ecosystems is linked with quality of life, mainly in highly urbanized regions. I suggest emphasizing how landscape dynamics could impact human health status, and why it is important to maintain a balance between natural and constructed areas. Of course, my suggestion implies the provision of a brief argumentation to increase the suitability of your paper within the journal's aims and scope.
Response 2: We have added the description of “how the quality of ecosystems is linked with quality of life” and “why it is important to maintain a balance between natural and constructed areas” in introduction (line 38-43). And we also cited reference of [3] Mushore et al. (2017) to explain “how landscape dynamics could impact human health status” (line 43-44).
Further I will provide specific comments on the manuscript's section and text:
Point 3: Abstract: I recommend rethinking the information you wish to provide in the abstract. This section is very important as it makes a reader reading the whole paper. Thus, I suggest to provide brief descriptions of the methods used and a glimpse to the major finding or findings of your paper. Also, in the end of your abstract you should provide an argument backing the originality and applicability provided by your study.
Response 3: We rewrote the abstract and added the argument at the end (line 12-30).
Point 4: Lines 14-15: The word "study" repeats and it alters the coherence of the sentence. I suggest: "[...] the analysis focused on Jiangsu Province in Eastern China as a case study are, [...]".
Response 4: We rewrote the abstract and modified this expression.
Point 5: Introduction: I suggest to clearly emphasize which are the study's main objectives at the end of this section. After that you should describe the methodologies and results in relation with these objectives. Otherwise the paper lacks a clear focus.
Response 5: We have added the objectives and the introduction of the methods and main results (line 85-98).
Point 6: Line 99: I suggest: "We have applied our methods using data recorded in the period between 2000 and 2015".
Response 6: We revised it as your suggestion (line 126).
Point 7: Line 103: Citation needed. Some readers may not know which is the land-use standard classification system.
Response 7: We cited the study of [8] Jin et al. (2019) to solve this problem (line 130-131).
Point 8: Line 112: I suggest: "[...] using ArcGIS 10.5 and Origin 9.0. softwares".
Response 8: We corrected it as your expression (line 139-140).
Point 9: 2.3. Methodologies: I recommend that for each index to clearly specify which are the required input data. In this way the methods are replicable by other researchers in their own case studies, fact that may increase your citation numbers and the journal's visibility.
Response 9: All indices were analyzed in the software ArcGIS 10.5 with using the same land-use data. The difference was just for choosing different processing tools. To avoid duplication, we have not added the input data for each index.
Point 10: Lines 114-115: Unclear, please consider rephrasing it to provide more clarity to the sentence.
Response 10: We have revised it to make it clearly (line 142-144).
Point 11: Lines 118-121: Unclear, please consider rephrasing it to provide more clarity to the sentence.
Response 11: We have revised it to make the expression more clearly (line 146-148).
Point 12: Lines 122-129: I suggest to reconsider the argumentation within this paragraph. In its current form it is unclear for the reader. Try using briefer and shorter sentences. I understand that you have calculated LVI and LHAI indexes at a county level, meaning you have 78 LVI and 78 LHAI values. Is that right?
Response 12: Your understanding is correct, we have 78 LVI and 78 LHAI values. We rewrote this paragraph (line 149-153).
Point 13: 2.3.1. Landscape Vulnerability Index
Line 142: I suggest :"land uses" to avoid repeating the word "types".
Response 13: We corrected it as your suggestion (line 166).
Point 14: Line 158: Which study?
Response 14: We added the study of [23] Song et al. (2015) (line 182-183).
Point 15: Line 163: How are PRD, SHDI and SHEI established? I reckon that they are calculated using spatial analysis tools and GIS processing. Right? However, a short description is required, to proper disseminate the methods to a wider range of readers.
Response 15: Yes, you are right. We added references and a short description (line 185-187).
Point 16: Line 165: the word "landscape" implies the fact that it refers to land, thus it is redundant to say "land landscape".
Response 16: We revised it (line 190).
Point 17: Line 169: The same as mentioned before.
Response 17: We revised it (line 194).
Point 18: Line 171: It is not clear what Ai and A stand for?
Response 18: We added the meaning (line 198).
Point 19: Table 1: These values were set empirical by the authors? If so, than you should provide more information on how the values were set, otherwise they lack the necessary scientific ground.
Response 19: These values are based on previous studies, we have written references before, but it may not be clear. So we changed the citation location of references (line 198-199).
Point 20: 2.3.3. Spatial Gravity center model: Without trying to be rude, I can't find any relevance of this analysis within your study. I'm not saying it is wrong but "cui prodest"? I understand that you wanted to emphasize how the pressure on landscape quality moved throughout the analyzed period with the region, but this can be observed from the maps as well. Also the way the methods and results are exposed are really fuzzy. If you wish to keep this analysis in your paper, I recommend providing more arguments to highlight why it is useful and what it shows. Link it to a paper objective (which you don't have at the moment).
Response 20: We have studied these comments carefully. Eventually we deleted LHAI’s center of gravity analysis because it doesn’t make much sense, but LVI’s center of gravity analysis was retained. The reason is as follows: We established the spatial gravity center model was to analyze the LVI change rate, not the value of LVI. So this content is different from what is shown in Figure 2. Firstly, we briefly introduced the object of the model in the introduction (line 85-88). We have comprehensively explained why it is useful to apply spatial gravity center model in Section2.3.3 (line 204-213). We have also added some explanation of the model’s establishment (line 276-278) and provided more details of “what it shows” (line 285-288). In addition, we referenced the study of [35] Balsa-Barreiro et al. (2019) to add arguments and enriched the discussion of the results of gravity central analysis model (line 297-302).
Point 21: Results-Figure 2: I recommend changing the colors used in this maps. You used different shades of green for low vulnerability and high vulnerability which may be confusing. I recommend using different shades of the same color for your scale, with an increasing tone from low to high.
Response 21: The Figure 2 was drawn again to make it clear (line 257).
Point 22: Lines 242-245: Please consider to divide the phrase into shorter and briefer sentences as in the current form it is hard to comprehend.
Response 22: We revised the phrase into shorter and briefer sentences (line 276-278).
Point 23: It is quite hard to read the map in its current form. It is hard to distinguish the circles from squares as they are too small and the colors hard to be identified. I recommend using larger symbols and different colors. Besides that, the not explained in the legend, such as the arrows. What does the colors represent? or the direction?
Response 23: We added some explanation of the map contains items and drew again the Figure 3 to make it bigger (line 282).
Point 24: Line 291: You should cite a figure before it is exposed to the reader. In your case is the other way around. Please address this issue.
Response 24: We adjusted the position of the Figure 5 (line 319).
Point 25: Line 295: Redundant. Please keep on "landscape" or "land use".
Response 25: This section belonged to the LHAI’s center of gravity analysis. Based on your 20th suggestion, we have deleted this section.
Point 26: Figure 6 and Table 3 must be revealed to the reader before passing to other subsections.
Response 26: This section belonged to the LHAI’s center of gravity analysis. Based on your 20th suggestion, we have deleted this section.
Point 27: Figure 6: The same issues emphasized for Figure 5 are occurring here. Please address them.
Response 27: This section belonged to the LHAI’s center of gravity analysis. Based on your 20th suggestion, we have deleted this section.
Point 28: Line 354: The Journal's recommendation, as well as the scientific writing customs indicates that a figure should be inserted after the ending of the paragraph in which it was first cited. It is not the case for your paper. Please address this issue and follow the above mentioned recommendations.
Response 28: We adjusted the position of the Figure 7 (line 402).
Point 29: Table 6: It is not clear, or the authors failed to provide sufficient information about what type of correlation was used for this analysis.
Response 29: We added some explanation to the correlation (line 393-398), and also provided the significance of each zone in Table 5 (Because of other changes, Table 5 in the revision is the Table 6 of the original text) (line 400).
Point 30: References: I have noticed that only 24% of your reference (24.48% more exactly) are not Chinese published papers, and from those, only one reference is from the last 5 years. While I personally acknowledge and appreciate the contribution of Chinese researchers within the international scientific community, I don't think that topics such as landscape dynamics are not tackled by others as well. The content of your references makes me wonder if your paper is not better suited for a Chinese national journal rather than an international one. Please address this issue.
Response 30: We have supplemented articles written by foreigners in relevant research areas. Now, 23 of the all 50 references are written by foreigners, and 65.22% are from the last 5 years.
Special thanks to you for your good comments.
We appreciate for Editors/Reviewers’ warm work earnestly, and hope that the correction will meet with approval.
Once again, thank you very much for your comments and suggestions.

Reviewer 3 Report
This paper analyzed the spatial and temporal characteristics of landscape fragility caused by human activities by constructing the LVI and LHAI. And the authors chose Jiangsu (China) as the case area and did some serious study, but I think this paper needs more improvements:
Introduction
1.The authors referred that "……many countries try to curb urban expansion by formulating land protection policies." But the authors just provided one country China as the example in the paper. Please add more examples (at least two examples). (Lines 36-39)
2.The authors wrote "The imbalance of economic levels in all counties has led to different development modes, so that human factors have become the main driving force for the fragility of regional land ecosystems." What are the different modes? Why the human factors are the main driving force? From the introduction, I cannot find the relationship between the imbalance of economic levels and human factors.
Materials and Methods
1.In the 2.2, the authors wrote " According to the land use standard classification system……"(line 103).
2. Is it a standard in China? Please add the reference or document to support this classification. In the 2.3.1, the authors normalize the Vi values. How did the authors normalize the values? What is the method? Please add the reference. (line 145)
3. In the 2.3.1, the authors wrote "According to the previous study, a=0.5, b=0.3, c=0.2"(line 158). But the reference [27] is about the oasis landscape. It is very different for the kind of landscape the author referred in this paper. Please provide the valid references.
Results and Discussions
1.How did the authors classify the vulnerability level in the study area? (lines 221-222) Please add the reference.
2. How did the authors divide the LHAI? (line 274-275) Please add the reference.
3. Table 6 (Page 14) is very hard to read and understand. Please make it easy for readers to understand.
4.This part is called "Results and Discussions", but I cannot find any part about "Discussion", please add the contents of "Discussion".
Conclusions
1.This paper can be used as a reference for the other cities in terms of methods, but different cities have different situations, and whether the methods can be copied is still questionable.
2. Isn't it any limitations in this paper? I think the authors need to add some limitations.
3.The conclusion part is not enough. The authors need to tell the readers the theoretical and practical contributions of this paper.
Author Response
Response to Reviewer 3 Comments
Dear Editors and Reviewers:
Thanks for your letter concerning our manuscript entitled “Coastal Landscape Vulnerability Analysis in Eastern China — Based on Land-Use Change in Jiangsu Province” (ID: ijerph-704770). Those comments are helpful for revising and improving our paper, as well as the important guiding significance to our researches. We have studied comments carefully and made correction which we hope meet with approval. Revised portion are marked in red in the paper. The main corrections in the paper and the responds to the reviewers’ comments are listed below point by point.
Authors’ Responses to Reviewer's Comments (Reviewer 3)
Author’s Notes
Reviewer 3:
This paper analyzed the spatial and temporal characteristics of landscape fragility caused by human activities by constructing the LVI and LHAI. And the authors chose Jiangsu (China) as the case area and did some serious study, but I think this paper needs more improvements:
Introduction
Point 1: The authors referred that "……many countries try to curb urban expansion by formulating land protection policies." But the authors just provided one country China as the example in the paper. Please add more examples (at least two examples). (Lines 36-39)
Response 1: We supplemented three examples from other countries with the reference of [4] Oliveira et al. (2019) (line 46-51).
Point 2: The authors wrote "The imbalance of economic levels in all counties has led to different development modes, so that human factors have become the main driving force for the fragility of regional land ecosystems." What are the different modes? Why the human factors are the main driving force? From the introduction, I cannot find the relationship between the imbalance of economic levels and human factors.
Response 2: We added explanation for these part in the introduction (line 75-83).
Point 3: In the 2.2, the authors wrote " According to the land use standard classification system……"(line 103). Is it a standard in China? Please add the reference or document to support this classification.
Response 3: The classification system is based on Resource and Environmental Science Data Center (RESDC) of the Chinese Academy of Sciences (http://www.resdc.cn/). We cited the study of [8] Jin et al. (2019) to support this classification (line 130-131).
Point 4: In the 2.3.1, the authors normalize the Vi values. How did the authors normalize the values? What is the method? Please add the reference. (line 145)
Response 4: We added some explanation to the method of normalization, and added the reference of [22] Tian et al. (2019) to support it (line 168-170).
Point 5: In the 2.3.1, the authors wrote "According to the previous study, a=0.5, b=0.3, c=0.2"(line 158). But the reference [27] is about the oasis landscape. It is very different for the kind of landscape the author referred in this paper. Please provide the valid references.
Response 5: We provided the valid references of [23] Song et al. (2015) and [22] Tian et al. (2019) to support it. The research area of the former is in Kunshan City and the latter is in Zhejiang Province. Kunshan City is within the boundaries of Jiangsu Province, and Zhejiang Province is adjacent to Jiangsu Province. Both of them have a similar land classification to Jiangsu Province (line 182-183).
Results and Discussions
Point 6: How did the authors classify the vulnerability level in the study area? (lines 221-222) Please add the reference.
Response 6: We added the references of [8] Jin et al. (2019) and [31] Li et al. (2017) (line 253-255).
Point 7: How did the authors divide the LHAI? (line 274-275) Please add the reference.
Response 7: We added the references of [8] Jin et al. (2019) and [31] Li et al. (2017) (line 315).
Point 8: Table 6 (Page 14) is very hard to read and understand. Please make it easy for readers to understand.
Response 8: We have provided more details of the explanation to the correlation (line 393-398), and also provided the significance of each zone in Table 5 (Because of other changes, Table 5 in the revision is the Table 6 of the original text) (line 400).
Point 9: This part is called "Results and Discussions", but I cannot find any part about "Discussion", please add the contents of "Discussion".
Response 9: The first submitted version has some part about "Discussion", which are in line 265-267, 332-335, 352-358, 387-391, 433-437, 444-449 and marked in blue but it may not be enough. We have added more content to enrich the "Discussion" (line 297-302, 362-372, 423-431, 437-442).
Conclusions
Point 10: This paper can be used as a reference for the other cities in terms of methods, but different cities have different situations, and whether the methods can be copied is still questionable.
Response 10: This sentence may not have been written accurately before, we have deleted it.
Point 11: Isn't it any limitations in this paper? I think the authors need to add some limitations.
Response 11: We added the limitations in the discussion (line 457-461).
Point 12: The conclusion part is not enough. The authors need to tell the readers the theoretical and practical contributions of this paper.
Response 12: We also rewrote the conclusions and added the contributions of this paper (line 477-489).
Special thanks to you for your good comments.
We appreciate for Editors/Reviewers’ warm work earnestly, and hope that the correction will meet with approval.
Once again, thank you very much for your comments and suggestions.

Round 2
Reviewer 2 Report
Dear authors,
Thank you for resubmitting an improved version of your paper. I also want to thank you for answering the previous review in a detailed fashion. I consider that your paper is much more comprehensive and worthy for publishing in IJERPH. The presented methodological approach is well described and it can provide an assessment model for other researchers and for other case studies. Therefore, I am happy to recommend the paper to accepted for publication.
Best regards and good luck in your future researches!
Author Response
Response to Reviewer 2 Comments
Dear Editors and Reviewers:
Thanks for your letter concerning our manuscript entitled “Coastal Landscape Vulnerability Analysis in Eastern China — Based on Land-Use Change in Jiangsu Province” (ID: ijerph-704770). Those comments and approval are helpful for improving our paper and increasing our passion for further research.
Authors’ Responses to Reviewer's Comments (Reviewer 2)
Author’s Notes
Reviewer 2:
Point 1:
Dear authors,
Thank you for resubmitting an improved version of your paper. I also want to thank you for answering the previous review in a detailed fashion. I consider that your paper is much more comprehensive and worthy for publishing in IJERPH. The presented methodological approach is well described and it can provide an assessment model for other researchers and for other case studies. Therefore, I am happy to recommend the paper to accepted for publication.
Best regards and good luck in your future researches!
Response 1: Thanks for your appreciation. It is with your suggestions that we can continually modify the paper and make it better. We appreciate your warm work earnestly. Once again, special thanks to you for your good comments and suggestions. Best regards!

Reviewer 3 Report
Thank you for the authors to modify your paper seriously. And I think this paper now is fit for the publication standard of Journal IJERPH. But before publication, I think there are still some modification in this paper.
Thank you for the authors to modify your discussion, but I suggest the authors need to add a new part which is called "Discussion" in this paper. The part of "Limitation" need to be put in the Conclusion part.Author Response
Response to Reviewer 3 Comments
Dear Editors and Reviewers:
Thanks for your letter concerning our manuscript entitled “Coastal Landscape Vulnerability Analysis in Eastern China — Based on Land-Use Change in Jiangsu Province” (ID: ijerph-704770). Those comments are helpful for improving our paper. We have studied comments carefully and made correction which we hope meet with approval. Revised portion are marked in red in the paper. The main corrections in the paper and the responds to the reviewers’ comments are listed below point by point.
Authors’ Responses to Reviewer's Comments (Reviewer 3)
Author’s Notes
Reviewer 3:
Thank you for the authors to modify your paper seriously. And I think this paper now is fit for the publication standard of Journal IJERPH. But before publication, I think there are still some modification in this paper.
Point 1: Thank you for the authors to modify your discussion, but I suggest the authors need to add a new part which is called "Discussion" in this paper.
Response 1: Thanks for your appreciation and suggestions, we have studied these comments carefully. After repeated thinking, we have not set the "Discussion" into a separate part, and still merged the "Discussion" and the "Results" into a unified part, called "Results and Discussion". The reasons are as follows: 1) The results of this article include various aspects, such as the annual and spatial changes of LVI, the annual and spatial changes of LHAI, the correlation analysis between LVI and LHAI at county and provincial scales, etc. A result followed by a corresponding discussion makes it easier for readers to understand the correspondence between the discussion and the result. 2) There is a sequential relationship between the discussion of different research contents. For example, the reasons for the different correlations between LVI and LHAI in different counties can be analyzed in Section3.4 only if the causes of LVI and LHAI changes in each county are discussed in Section3.1 and Section3.2. Then we can discuss the future sustainable land development policies of the counties based on the zoning results of Section3.4. In other words, the research content of the latter is based on the discussion of the former, so from the logical consideration of the paper, it is necessary to discuss directly after each result is obtained. 3) Many papers have been published in Journal IJERPH with the "Results" and "Discussions" as merged parts called "Results and Discussion". Examples are as follows:
- Liu, M.; Han, G.; Zhang, Q., Effects of Soil Aggregate Stability on Soil Organic Carbon and Nitrogen under Land Use Change in an Erodible Region in Southwest China. International Journal of Environmental Research and Public Health 2019, 16, (20).
- Alaimo, M. G.; Varrica, D., Recognition of Trace Element Contamination Using Ficus macrophylla Leaves in Urban Environment. International Journal of Environmental Research and Public Health 2020, 17, (3).
- Maftei, A. E.; Buzatu, A.; Buzgar, N.; Apopei, A. I., Spatial Distribution of Minor Elements in the Tazlau River Sediments: Source Identification and Evaluation of Ecological Risk. International Journal of Environmental Research and Public Health 2019, 16, (23).
These papers are well structured and has great reading value. For the above reasons, there is no separate "Discussion" part. We appreciate for your good suggestions, and hope that our explanation will meet with approval.
Point 2: The part of "Limitation" need to be put in the Conclusion part.
Response 2: We revised it as your suggestion (line 472-477).
Once again, thank you very much for your warm work earnestly and suggestions.
Best regards!
